# Associations between Insomnia, Daytime Sleepiness, and Depressive Symptoms in Adolescents: A Three-Wave Longitudinal Study

**DOI:** 10.3390/jcm11236912

**Published:** 2022-11-23

**Authors:** Xianchen Liu, Yanyun Yang, Zhenzhen Liu, Cunxian Jia

**Affiliations:** 1Center for Public Health Initiatives, University of Pennsylvania, Philadelphia, PA 19104, USA; 2Department of Educational Psychology and Learning Systems, Florida State University, Tallahassee, FL 32306, USA; 3School of Psychology, Northeast Normal University, Changchun 130024, China; 4Department of Epidemiology, School of Public Health, Cheeloo Medical College, Shandong University, Jinan 250012, China

**Keywords:** daytime sleepiness, insomnia, depression, adolescence, longitudinal study

## Abstract

Background: Insomnia, daytime sleepiness, and depressive symptoms are prevalent in adolescents. This three-wave prospective study examined the associations between the three symptoms in adolescents. Methods: A total of 6995 schoolchildren in 7th and 10th grades (Mean age = 14.86 years) participated in a longitudinal study of behavior and health in Shandong, China. Standardized rating scales were used to assess symptoms of insomnia, daytime sleepiness, and depression in November–December in 2015, 1 year later, and 2 years later. Results: Insomnia was cross-sectionally associated with 10–14-fold increased odds of daytime sleepiness and 5–9-fold increased odds of depression. Daytime sleepiness was associated with 4–5-fold increased odds of depression. Insomnia, daytime sleepiness, or depression at a later time point was significantly predicted by itself at earlier time points. Insomnia was a significant predictor of daytime sleepiness and depression and a mediator between depression and daytime sleepiness. Daytime sleepiness was a significant predictor of insomnia and a mediator between depression and insomnia. Depression was a significant predictor of insomnia and daytime sleepiness and a mediator between insomnia and daytime sleepiness. Conclusions: Insomnia, daytime sleepiness, and depressive symptoms were highly comorbid in adolescents. The associations of insomnia with daytime sleepiness and depression were bidirectional. Depression predicted daytime sleepiness, but not vice versa. Further research is needed to understand the underlying neurobiological mechanisms between insomnia, daytime sleepiness, and depression during adolescence.

## 1. Introduction

Adolescents experience significant sleep changes and are vulnerable to sleep problems due to the interaction of endogenous circadian rhythms, pubertal development, increased social, academic, and extracurricular demands, decreased parental monitoring, and excessive digital media use [1,2,3]. Insufficient sleep, irregular sleep patterns, and daytime sleepiness (DS) are prevalent in adolescents [4,5,6,7]. Adolescents also experience rapid emotional and social changes with increased risk of behavioral and emotional problems [8,9]. For example, about a third of adolescents have depressive symptoms and 20% suffer from major depressive disorder during their lifetime [10]. Insomnia, DS, and depression all have negative impacts on individuals’ daily functioning and quality of life and increase risk of substance use, accidents, and suicidal behavior [1,11,12].

Growing evidence has demonstrated a bi-directional relationship between insomnia and depression such that insomnia can increase risk of depression and vice versa [13,14,15]. Recent epidemiological studies have shown that, compared to short sleep duration, DS has a stronger association with poor daytime functioning [16,17,18]. DS is a mediator between insomnia and poor academic performance [19] and a significant predictor of depression [20] and suicidal behavior [11]. On the other hand, depression increases risk of DS [21]. Persistent and incident insomnia increases risk of DS [22]. DS may also be a cause of insomnia due to too much sleep during the day and/or circadian rhythms disturbance. Therefore, the relationships between insomnia, DS, and depression may be of mutual causation, where the three problems are each a cause and/or a mediator of the others. To date, however, little empirical work has investigated the interplays between insomnia, DS, and depression in multi-wave longitudinal studies in adolescents [23].

In a recent longitudinal study of adolescents [23], the authors collected three waves of data 1 year apart over 2 years and found that variability in sleep duration and efficiency and DS predicted subsequent anxiety and depressive symptoms. The bidirectionality was more likely to be observed for DS compared with other sleep variables. However, participants were a small convenient sample (*n* = 246). The study did not examine the mediating effects of one sleep or mental health variable on the relationship between other variables. Gaining a better understanding of the prospective associations between insomnia, DS, and depression in large community sample of adolescents may have important implications for early detection and intervention of sleep disturbance and depression to prevent negative health outcomes for at-risk adolescents.

Using the large three-wave longitudinal data collected 1 year apart for 2 years (*n* = 6995), the current analysis aimed to (1) estimate the comorbidity of insomnia, DS, and depressive symptoms in Chinese adolescents and (2) examine the associations between insomnia, DS, and depressive symptoms over two years. We hypothesized that (1) symptoms of insomnia, DS, and depression are highly comorbid in adolescents; (2) insomnia, DS, and depression are predicted by the same variable measured at earlier time points; and (3) insomnia, DS, and depression have bidirectional relationships—each is predicted by the other two variables measured at an earlier time point. Our exploratory objective was to determine whether insomnia, DS, or depression mediates the association between the other two variables. There exist age and gender differences in sleep and mental health problems in children and adolescents [5,24,25,26]. However, little is known if the longitudinal relationships of insomnia, DS, and depression differ by age and gender. We thus further explored whether these relationships between the three symptoms are moderated by gender and stage of adolescence. Early adolescence was indicted by 7th graders (freshmen in middle school, mean age = 12.81 years) and middle adolescence by 10th graders (freshmen in high school, mean age = 15.77 years) at baseline [27].

## 2. Materials and Methods

### 2.1. Participants and Procedure

Data for this prospective analysis were derived from the Shandong Adolescent Behavior and Health Cohort (SABHC). The SABHC is a prospective study with 3 waves of data collection occurred 1 year apart from 2015 to 2017 in Shandong, China. Detailed sampling and data collection procedure were described in previous publications [28,29,30]. Briefly, 11,831 students from five middle schools (7th to 9th graders) and three high schools (10th to 12th graders) in Shandong participated in the baseline survey in November-December 2015 (T1). Due to graduation, only 7th and 10th graders at baseline (*n* = 6995) were resurveyed one year later (T2, *n* = 5807, response rate = 83.0%) and two years later (T3, *n* = 4853, response rate = 69.4%).

At each wave, participants were asked to fill out a self-administered questionnaire in their classrooms during typical school hours. It took approximately 45 min to complete the questionnaire. Participation in the study was voluntary. There were no penalties or consequences of any kind if students did not want to participate. The SABHC was approved by the research ethics committee of Shandong University School of Public Health. Informed consent was obtained from participants, and permission was obtained from parents, class teachers, and principals of participating schools.

### 2.2. Measures

#### 2.2.1. Insomnia Symptoms

The 8-item Youth Self-Rating Insomnia Scale (YSIS) [31] was used to measure insomnia symptoms in the past month. Each item is rated on a 5-point scale ranging from 1 to 5. Example items are “During the past month, how often would you say you have had trouble falling asleep?” “During the past month, how would you rate the quality of your sleep overall?” The scale score was computed by summing the item scores, with a higher score indicating a greater insomnia severity. A total score ≥ 26 was used to define clinically relevant insomnia symptoms [31]. The coefficient alpha of the scale with the current sample was 0.80, 0.81, and 0.83 at T1, T2, and T3, respectively.

#### 2.2.2. Daytime Sleepiness

The Chinese adolescent daytime sleepiness scale (CADSS) was used to assess daytime sleepiness [32]. The CADSS consists of 7 questions that ask about individual’s general feeling of drowsiness and dozing off at different situations during the daytime in the past month. Example items are “During the past month, how often would you say you feel sleepy during the day?” “During the past month, how often would you say you have dozed off in the morning classes?” All 7 items are rated on a Likert scale from 1 = Never to 5 = Almost every day. Summing up the item scores yields a total CADSS score, ranging from 7 to 35. A higher CADSS score indicates a greater daytime sleepiness. A total score ≥ 23 was used to identify clinically relevant daytime sleepiness [32,33,34]. The coefficient alpha with the current sample was 0.90, 0.91, and 0.91 at T1, T2, and T3, respectively.

#### 2.2.3. Depressive Symptoms

The 20-item Epidemiological Studies Depression Scale (CES-D) [35] was used to measure depressive symptoms. Participants were asked to rate how often they experienced each symptom in the past seven days: 0 = “less than 1 day”, 1 = “1–2 days”, 2 = “3–4 days”, and 3= “5–7 days”. A total score was calculated by summing the item scores, with higher scores indicating higher severity of depressive symptoms. A cut-off score of 30 was used to define clinically relevant depression in Chinese adolescents [36]. The coefficient alpha of the scale with the current sample was 0.86, 0.85, and 0.88 at T1, T2, and T3, respectively.

#### 2.2.4. Covariates

Based on the literature [1,7,26,37,38], the following covariates were included in the analysis: gender, grade level (7th/10th), chronic disease (yes/no), ever cigarette smoking (yes/no), ever alcohol consumption (yes/no), nocturnal sleep duration on weekdays, father’s education (primary school, middle school, high school, professional school, or college and above), and perceived family economic status (excellent, good, fair, poor, or very poor). Father’s education was included as a covariate because family social economic status is more likely to be determined by father’s education than mother’s education in traditional Chinese culture and because mother’s education was significantly correlated to father’s education (r = 0.55). All these variables were self-reported at T1 except nocturnal sleep duration on weekdays as a time-varying covariate at T1, T2, and T3.

### 2.3. Statistical Analysis

Statistical analyses followed five steps. First, chi-square tests were conducted to test whether adolescent and family factors measured at baseline were similar for adolescents who participated in baseline survey, 1-year follow-up, and 2-year follow-up. Second, prevalence and comorbidity rates of insomnia, DS, and depressive symptoms at each time point were computed. Logistic regressions were then performed to show the comorbid associations between the three symptoms. Third, bi-correlation coefficients between the three symptoms were computed. Fourth, cross-lagged panel model was performed to examine their longitudinal associations over the three time points. Figure 1 shows the hypothesized relationships. Adolescent and family covariates were included in the panel model to control for their effects on the associations between the three symptoms. Fifth, two-group cross-lagged panel models were used to examine moderation effects of gender (male vs. female) and grade level (7th vs. 10th) on the longitudinal associations between the three symptoms. Models were considered a good fit if χ^2^ test statistic was not significant, comparative fit index (CFI) ≥ 0.95, root mean square error of approximation (RMSEA) ≤ 0.05, and standardized root mean square residual (SRMR) ≤ 0.06 [39]. However, due to large sample size, model-data fit evaluation was mainly based on CFI, RMSEA, and SRMR. Mediation effects were computed as the product of cross-lagged path coefficients and were tested for significance using Sobel test [40]. Cross-lagged analyses were conducted in Mplus version 8.7(Muthén & Muthén, Los Angeles, CA, USA) [41]. All other statistical analyses were performed using IBM SPSS Statistics for Windows, Version 27.0 (Armonk, NY, USA: IBM Corp.).

## 3. Results

### 3.1. Sample Characteristics

Of the 6995 participants, 51.4% were male, 31.3% were 7th graders, and 68.7% were 10th graders. Mean ages at baseline were 14.86 years (SD = 1.50) for the entire sample, 12.81 (SD = 0.58) for 7th graders, and 15.77 (SD = 0.66) for 10th graders. Mean nocturnal sleep duration on weekdays was 7.20 hrs (SD = 2.76); 59.6% slept < 7 h per night. The proportions of adolescents who reported having chronic disease, ever cigarette smoking, ever alcohol consumption, and poor family economic statues at baseline were 4.0%, 3.5%, 7.3%, and 12.3%, respectively. Father’s education was high school or below in majority of adolescents (82.3%).

A total of 5807 (83.0%) and 4853 (69.4%) participants completed 1-year and 2-year follow-up surveys, respectively. The major reason for loss to follow-up was that students transferred to other schools/classes or were absent from school on the day of the survey. Chi-square tests revealed that 10th graders (χ^2^ = 78.75, *p* < 0.01), adolescents who reported having ever cigarette smoking (χ^2^ = 35.52, *p* < 0.01) and alcohol consumption (χ^2^ = 24.45, *p* < 0.01) at baseline were less likely to participate in the follow-up surveys.

### 3.2. Prevalence and Comorbidity of Insomnia, Daytime Sleepiness, and Depressive Symptoms

As shown in Figure 2, the prevalence rates of clinically relevant symptoms of insomnia, daytime sleepiness (DS), and depression were 19.4%, 25.3%, and 12.0% at T1, 16.7%, 25.3%, and 7.1% at T2, and 16.0%, 25.1%, and 11.5% at T3, respectively. Approximately 11–12% of adolescents had both insomnia and DS, 4–6% had both insomnia and depressive symptoms, 4–6% had both DS and depressive symptoms, and 3–4% had all three symptoms.

Table 1 presents the cross-sectional associations between the three symptoms at T1, T2, and T3. Insomnia was associated with 10–14-fold increased odds of DS and 5–9-fold increased odds of depression. DS was associated with 4–5-fold increased odds of depression. For instance, the odds ratios of depression associated with DS at T1, T2, and T3 were 4.57 (95%CI = 3.92–5.32), 5.44 (95%CI = 4.40–6.72), and 4.91 (95% = 4.08–5.91), respectively.

### 3.3. Panel Models across Three Time Points

Table 2 presents bi-correlations and descriptive statistics of insomnia, DS, and depressive symptoms over the three time points. The three symptoms were significantly and positively correlated to each other (all *p* < 0.001), ranging from 0.27 to 0.69. The correlations of the same variable over time were moderately high (0.48–0.59 for insomnia, 0.44–0.52 for DS, and 0.46–0.57 for depressive symptoms). All variables were positively skewed and did not follow normal distributions. Therefore, robust maximum likelihood estimation method was used to test the cross-lagged panel model.

The model demonstrated an adequate fit to the data (see Table 3). Figure 1 reports standardized path coefficients after adjusting for adolescent and family covariates. The results were summarized below.

Insomnia, DS, and depression at T1 positively and significantly (all *p* < 0.01) predicted the same symptoms at T2 and T3. Specifically, the autoregressive coefficients for adjacent time points (T1→T2 and T2→T3) were 0.36–0.39 for insomnia, 0.31–0.32 for DS, and 0.41–0.43 for depression, respectively. The autoregressive coefficients from T1→T3 were 0.18, 0.17, and 0.21 for insomnia, DS, and depression, respectively.Insomnia and DS had a bidirectional relationship. Specifically, the cross-lagged coefficients from insomnia(T1)→DS(T2) and DS(T1)→insomnia(T2) were 0.11 (*p* < 0.01) and 0.05 (*p* < 0.01), respectively. The corresponding coefficients from T2→T3 were 0.09 (*p* < 0.01) and 0.07 (*p* < 0.01), respectively.The relationship between insomnia and depression was bidirectional, as evidenced by the significant cross-lagged coefficients of 0.12 (*p* < 0.01) and 0.09 (*p* < 0.01) from depression(T1)→insomnia(T2) and insomnia(T1)→depression(T2), and significant coefficients of 0.09 (*p* < 0.05) and 0.07 (*p* < 0.05) from depression(T2)→insomnia (T3) and insomnia (T2)→depression(T3), respectively.The relationship between DS and depression was unidirectional, with depression at an earlier year predicting DS at the follow-up year. The cross-lagged coefficients were 0.09 (*p* < 0.01) and 0.06 (*p* < 0.01) from depression(T1)→DS(T2) and depression(T2) →DS(T3), respectively. However, the cross-lagged coefficients from DS(T1)→depression(T2) and DS(T2) →depression(T3) were not statistically significant (*p* > 0.05).Four significant mediation effects were obtained: from depression(T1) →insomnia (T2)→daytime sleepiness(T3), from depression(T1)→DS(T2)→insomnia(T3), from insomnia(T1)→depression(T2)→DS(T3), and from DS(T1)→insomnia(T2)→depression(T3). The corresponding mediation effects were 0.010 (*p* < 0.01), 0.006 (*p* < 0.01), 0.006 (*p* < 0.01), and 0.004 (*p* < 0.05), respectively.

Two sets of two-group cross-lagged panel models were conducted to test moderation effects of gender (male vs. female) and grade level (7th vs. 10th grade), respectively, on the longitudinal associations among insomnia, DS, and depression. Table 3 shows model-data fit information for each two-group panel model. All four models yielded an adequate fit. More importantly, both chi-square difference tests were not significant (*p* > 0.01), suggesting that there was no significant gender difference and grade difference in the longitudinal associations among insomnia, daytime sleepiness, and depression.

## 4. Discussion

To our knowledge, this is the first study with measures at three time points over a period of 2 years to report the associations between symptoms of insomnia, daytime sleepiness (DS), and depression in a large sample of Chinese adolescents. The major findings are (1) insomnia, DS, and depressive symptoms were highly comorbid in adolescents; (2) the associations of insomnia with DS and depression were prospectively bidirectional; (3) depression predicted DS and was a mediator between insomnia and DS; (4) insomnia mediated the association between depression and DS; and (5) DS mediated the associations between depression and insomnia. As expected, insomnia, DS and depressive symptoms were prevalent and highly comorbid in adolescents [22,42,43,44]. The following discussion is focused on the prospective directionality and mediating effects between symptoms of insomnia, DS, and depression.

Consistent with previous studies [13,14,15,45,46], our longitudinal data analysis demonstrated that insomnia and depressive symptoms was bidirectional. The bidirectionality can be explained by the shared genetic liability and neurobiological mechanisms, such as an activation of inflammatory pathways, altered hypothalamic–pituitary–adrenal axis activity, disrupted neuroplasticity, REM sleep dysregulation, and disrupted circadian rhythm [47,48].

Although little empirical work has prospectively investigated the bidirectional relationship between insomnia and DS, the bidirectionality is expected. This is because night sleep disturbance can result in sleep insufficiency and/or use of sleep medication [44], which in turn may lead to daytime sleepiness. On the other hand, DS may cause sleeping too much during the day and increase the use of caffeinated beverages/stimulants, all of which can disrupt night sleep, and thereby result in difficulty initiating sleep and difficulty maintaining sleep. Comorbid conditions such as sleep apnea and circadian rhythm sleep disorders may be the other factors that contribute to the insomnia-DS bidirectional link.

Cross-sectional studies have shown that DS is related to depression in adolescents or young adults [49,50]. However, no causal relationship could be inferred from cross-sectional analyses. Longitudinal data on the association between DS and depression in adolescents are limited. In a three-wave longitudinal study over 2 years in 246 adolescents [23], the authors found that more depressive symptoms at age 16 predicted higher levels of sleepiness at age 17 while greater sleepiness at age 17 predicted more depressive symptoms at age 18. In the current longitudinal study, we found that depressive symptoms significantly predicted DS, but not the vice versa. There are two plausible explanations for depressive symptoms predicting DS. First, depression may compromise circadian rhythms and disrupt sleep, and this contributes to daytime sleepiness. Second, DS may be the side effects of antidepressants and sleep pills commonly used for depressive or insomnia symptoms. Given inconsistent findings in the limited studies, further research is warranted to investigate whether there is a bidirectional relationship between DS and depression. If the relationship is unidirectional, further research is also needed to confirm whether depression predicts DS or vice versa.

Examinations of mediation effects of insomnia, DS, or depressive symptoms on the relationships between two of the three symptoms are scarce. In the three-wave longitudinal data analysis, we found that insomnia partially mediated the association between depression and DS while DS mediated the associations between depression and insomnia. That is, depression is associated with subsequent DS partially via insomnia, and depression is associated with subsequent insomnia partially via DS. We did not have specific hypotheses/explanations on these mediation relationships as the analyses were exploratory. However, these mediation relationships may help understand and guide new research to determine the complicated relationships between insomnia, daytime sleepiness, and depression during the at-risk transition period from childhood to adulthood.

According to the literature, symptoms of insomnia, DS, and depression are more prevalent in high school students or middle adolescence students than in middle school peers [5,6,51,52,53]. Girls are more likely to report symptoms of insomnia, DS, and depression with advancing age during adolescence. However, little is known whether the prospective relationships between insomnia, DS, and depression vary by gender and age. The current three-wave panel model analysis showed that the associations between insomnia, DS, and depression were similar between males and females and between 7th and 10th graders—that is, there are no significant moderation effects of age and gender on the prospective relationships between insomnia, DS, and depressive symptoms. The finding may be of particular relevance and informs that prevention and intervention of insomnia, daytime sleepiness, and depression should start at early adolescence regardless of gender.

This study has several strengths, including large sample size, three time points of data collection, and standardized rating scales used to measure insomnia, DS, and depressive symptoms. Cross-lagged panel models, which were conducted within the structural equation modeling framework, have the advantages for examining causal relationships with longitudinal data while controlling for the effects of same variables measured at earlier time points and the effects of covariates.

The findings need to be interpreted with consideration of the following limitations. First, all data were collected based on self-report, which may lead to shared recalling biases. However, self-report is the method used to assess perceived severity of insomnia, DS, and depressive symptoms, especially in large-scale epidemiological studies. Second, insomnia, DS, and depressive symptoms were assessed using screening scales rather than clinical interview or diagnosis. Third, insomnia, DS, and depressive symptoms were all assessed 1 year apart. The effects of the changes of these symptoms between assessments could not be adjusted. Fourth, although multiple adolescent and family variables were statistically controlled, other variables, including circadian factors, sleep and mental disorders and their treatments could not be included in the analysis. For example, chronic pain [54,55] and sleep apnea and bruxism [56,57] that can cause insomnia, DS, and depressive symptoms were not statistically controlled and might have confounding effects on the associations between the three symptoms under investigation. Fifth, although chronic diseases as a dichotomous variable (i.e., yes or no) were included in the analysis, detailed information about the diseases (e.g., neurological diseases, past injuries, and psychiatric disorders) and their histories are unknown. Furthermore, although the sample size is large, the findings from Chinese adolescents may not be generalized to other adolescent populations because sleep and mental health problems in pediatric populations are influenced by psychosocial and cultural factors.

## 5. Conclusions, Clinical Implications, and Future Directions

The present study included a large sample of community adolescents, employed a rigorous design with three waves of data collection, and provided novel insights into the prospective relationships between symptoms of insomnia, daytime sleepiness, and depression. The findings of the current study suggest that insomnia, daytime sleepiness, and depression not only often co-occur but also have complicated associations. As expected, the associations of insomnia with DS and depressive symptoms were prospectively bidirectional. Depression predicted DS, but not vice versa. Insomnia mediated the association between depression and DS while DS mediated the association between depression and insomnia. Furthermore, the complex relationships of insomnia, DS, and depressive symptoms did not vary across age and gender.

These findings have important clinical implications. Insomnia, daytime sleepiness, and depression should be evaluated together in the routine clinical practice. Clinical interventions, including non-pharmacological treatments, such as cognitive behavioral therapy, should take the three comorbid conditions and their reciprocal associations into consideration for effective treatment of sleep disturbance and depression in adolescents.

Further research is needed to understand the underlying neurobiological mechanisms between insomnia, daytime sleepiness, and depression during adolescence. Clinical intervention research is needed to evaluate whether pharmacological and/or non-pharmacological therapy targeting insomnia, daytime sleepiness, and depression together in adolescent patients is more effective than targeting one or two of the three symptoms.

## Figures and Tables

**Figure 1 jcm-11-06912-f001:**
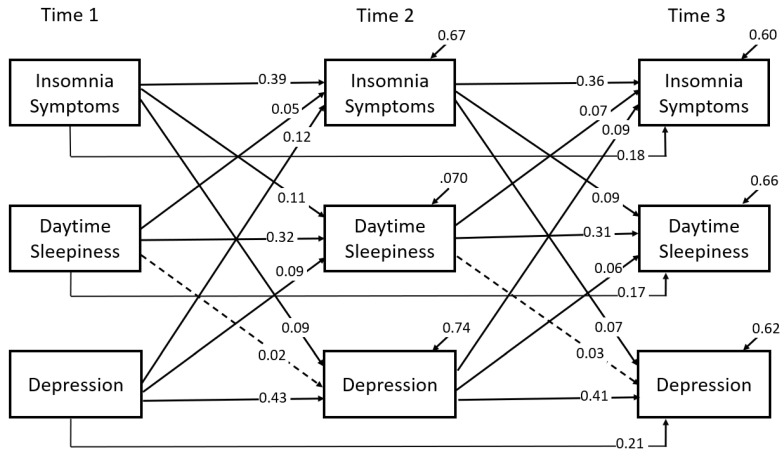
Standardized path coefficients from the cross-lagged panel model. Note. All path coefficients had *p* < 0.01 except those indicated by dashed lines (*p* > 0.05). Although not reported, covariances among variables at Time 1 and residual covariances at Time 2 and at Time 3 were freely estimated.

**Figure 2 jcm-11-06912-f002:**
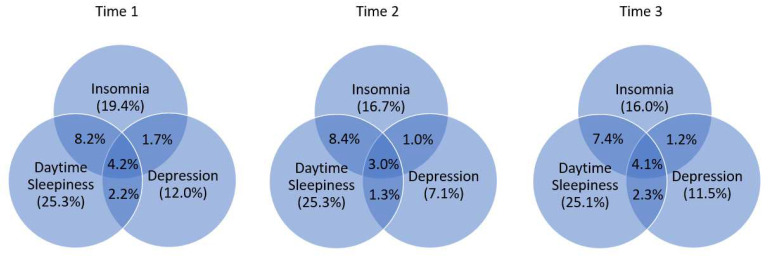
Comorbidities of insomnia, daytime sleepiness, and depressive symptoms at baseline (Time 1), 1 year later (Time 2), and 2 years later (Time 3) among adolescents.

**Table 1 jcm-11-06912-t001:** Cross-sectional associations between symptoms of insomnia, daytime sleepiness, and depression at baseline (Time 1), 1 year later (Time 2), and 2 years later (Time 3).

	Associations (OR, 95%CI)	
	Insomnia Symptoms	Daytime Sleepiness
**Time 1**		
Insomnia symptoms	1.00	
Daytime sleepiness	10.12 (8.82–11.60)	1.00
Depressive symptoms	5.84 (4.99–6.82)	4.57 (3.92–5.32)
**Time 2**		
Insomnia symptoms	1.00	
Daytime sleepiness	11.43 (9.75–13.39)	1.00
Depressive symptoms	8.81 (7.10–10.92)	5.44 (4.40–6.72)
**Time 3**		
Insomnia symptoms	1.00	
Daytime sleepiness	14.25 (11.86–17.12)	1.00
Depressive symptoms	6.87 (5.66–8.38)	4.91 (4.08–5.91)

**Table 2 jcm-11-06912-t002:** Bicorrelations and descriptive statistics of insomnia, daytime sleepiness, and depressive symptoms.

	1	2	3	4	5	6	7	8	9
**Time 1**									
1. Insomnia symptoms	1								
2. Daytime sleepiness	0.68	1							
3. Depressive symptoms	0.49	0.44	1						
**Time 2**									
4. Insomnia symptoms	0.54	0.42	0.37	1					
5. Daytime sleepiness	0.44	0.51	0.31	0.68	1				
6. Depressive symptoms	0.32	0.27	0.49	0.48	0.37	1			
**Time 3**									
7. Insomnia symptoms	0.48	0.39	0.32	0.59	0.46	0.38	1		
8. Daytime sleepiness	0.39	0.44	0.27	0.45	0.52	0.28	0.69	1	
9. Depressive symptoms	0.33	0.27	0.46	0.40	0.32	0.57	0.52	0.43	1
Mean	19.62	17.05	17.30	18.75	17.04	14.61	18.47	17.15	16.27
SD	6.31	7.64	9.85	6.53	7.84	8.98	6.64	7.94	10.47
Skewness	0.21	0.60	0.83	0.34	0.64	1.23	0.38	0.61	0.93
Kurtosis	−0.54	−0.56	0.57	−0.44	−0.52	2.05	−0.63	−0.53	0.80

All correlation coefficients had *p* < 0.001.

**Table 3 jcm-11-06912-t003:** Model-data fit information from cross-lagged panel models.

	χ^2^	*df*	CFI	RMSEA	SRMR	Δχ^2^	Δ*df*
**Overall sample**	159.21 *	24	0.994	0.028	0.016		
**By gender**							
Unconstrained model	180.33 *	48	0.994	0.028	0.018		
Constrained model	204.93 *	69	0.994	0.024	0.019	29.50	21
**By grade level (7th vs. 10th)**							
Unconstrained model	186.19 *	48	0.992	0.029	0.023		
Constrained model	214.85 *	69	0.992	0.025	0.024	34.18	21

* *p* < 0.01. Δχ^2^ indicated chi-square difference test statistic between unconstrained and constrained models. Both χ^2^ and Δχ^2^ referred to test statistics with robust correction because robust maximum likelihood estimation method was used for analyses.

## Data Availability

Data is available on request from the corresponding author.

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
