# Peer review of "Associations between Insomnia, Daytime Sleepiness, and Depressive Symptoms in Adolescents: A Three-Wave Longitudinal Study"

_jcm, 2022, doi:10.3390/jcm11236912_

Round 1
Reviewer 1 Report
The paper presented to me for review deals with the well-known but poorly proven issue of associations between insomnia, daytime sleepiness, and depressive symptoms in adolescents. Although the relationships seem obvious and are observed in clinical practice there is still a lack of large and reliable studies that confirm them in long-term observation.
The paper is based on a very large sample of respondents. The study is well designed and the results are presented clearly and lucidly. Undoubtedly, the work brings significant new knowledge to the field.
However, I have a few comments to add:
1. insomnia and daytime sleepiness and depressive/anxiety disorders are quite common in the general population but are also a symptom of various neurological, psychiatric and systemic diseases; this should be mentioned in the discussion when mentioning these potential groups of patients with common diseases like migraine (https://pubmed.ncbi.nlm.nih.gov/34073933/) or sleep apnea and bruxism (https://pubmed.ncbi.nlm.nih.gov/32160378/)
2. in limitations, it would be appropriate to mention that the exact medical history (including neurological diseases, past injuries, etc. and psychiatric diseases) of respondents is unknown
Author Response
Thank you for your comments. Attached please find our responses.

Reviewer 2 Report
Dear Authors,
Your article «Associations between insomnia, daytime sleepiness, and depressive symptoms in adolescents: A 3-wave longitudinal study» is a very interesting and useful for pediatrics practice. Really, adolescents experience significant sleep changes and are vulnerable to sleep problems due to the interaction of exogenous and endogenous factors, as well as increased risk of depressive disorder. But there aren't any large longitudinal studies to evaluate interaction between insomnia, daytime sleepiness and depression in adolescence. Your contribution to this field is need and valuable for pediatric health care. This research gaining a better understanding of the prospective associations between symptoms and conditions in community samples of adolescents may have important implications for early detection and intervention of sleep disturbance and depression to prevent negative health outcomes for at-risk adolescents. At the same time, some comments should be corrected and I would like to get answers to some questions:
1. The abstract does not indicate that the study included schoolchildren of the 7th grade, only the 10th?
2. What classification of adolescence did you use?
3. Why did you include the education of the father and not the mother, or both parents, in the covariates?
4. There are differences in the presentation of the material: In lines 186-189 (section 3.2) and in table 1, the approximate frequency of present of both insomnia and daytime sleepiness is 11-12%, insomnia and depression is 4-6%, daytime sleepiness and depression is 4-6%, and a combination of 3 symptoms is 3-4 %. But Figure 1 presents other data (7-8%, 1-1.7% and 1-2%, respectively). What data is considered correct? Edits should be made. Table 1 can be excluding, because it duplicates material in the text.
Author Response

(The authors gave the same response as above.)
